# Metabolic Profile of *Scytalidium parasiticum*-*Ganoderma boninense* Co-Cultures Revealed the Alkaloids, Flavonoids and Fatty Acids that Contribute to Anti-Ganoderma Activity

**DOI:** 10.3390/molecules25245965

**Published:** 2020-12-16

**Authors:** Rafidah Ahmad, Choon Kiat Lim, Nurul Fadhilah Marzuki, Yit-Kheng Goh, Kamalrul Azlan Azizan, You Keng Goh, Kah Joo Goh, Ahmad Bazli Ramzi, Syarul Nataqain Baharum

**Affiliations:** 1Metabolomics Research Laboratory, Institute of Systems Biology (INBIOSIS), Universiti Kebangsaan Malaysia, UKM, Bangi, Selangor 43600, Malaysia; fida.ahmad@ukm.edu.my (R.A.); kamalrulazlan@ukm.edu.my (K.A.A.); bazliramzi@ukm.edu.my (A.B.R.); 2Advanced Agriecological Research Sdn Bhd, No. 11 Jalan Teknologi 3/6, Taman Sains Selangor 1, Kota Damansara, Petaling Jaya, Selangor Darul Ehsan 47810, Malaysia; cklim@twinarrow.com.my (C.K.L.); nf91fadh@gmail.com (N.F.M.); gohykheng@aarsb.com.my (Y.-K.G.); gohyk@aarsb.com.my (Y.K.G.); gohkj@aarsb.com.my (K.J.G.)

**Keywords:** anti-Ganoderma, metabolomics, *Scytalidium parasiticum*, *Ganoderma boninense*, LC-TOF-MS analysis, biological control

## Abstract

In solving the issue of basal stem rot diseases caused by Ganoderma, an investigation of *Scytalidium parasiticum* as a biological control agent that suppresses Ganoderma infection has gained our interest, as it is more environmentally friendly. Recently, the fungal co-cultivation has emerged as a promising method to discover novel antimicrobial metabolites. In this study, an established technique of co-culturing *Scytalidium parasiticum* and *Ganoderma boninense* was applied to produce and induce metabolites that have antifungal activity against *G. boninense*. The crude extract from the co-culture media was applied to a High Performance Liquid Chromatography (HPLC) preparative column to isolate the bioactive compounds, which were tested against *G. boninense*. The fractions that showed inhibition against *G. boninense* were sent for a Liquid Chromatography-Time of Flight-Mass Spectrometry (LC-TOF-MS) analysis to further identify the compounds that were responsible for the microbicidal activity. Interestingly, we found that eudistomin I, naringenin 7-O-beta-D-glucoside and penipanoid A, which were present in different abundances in all the active fractions, except in the control, could be the antimicrobial metabolites. In addition, the abundance of fatty acids, such as oleic acid and stearamide in the active fraction, also enhanced the antimicrobial activity. This comprehensive metabolomics study could be used as the basis for isolating biocontrol compounds to be applied in oil palm fields to combat a *Ganoderma* infection.

## 1. Introduction

Malaysia has been known as one of the world’s top oil palm producers and exporters since the 1960s, and oil palm production has grown from a humble crop industry to become one of the most significant contributors to Malaysia’s gross domestic product. This industry is a booming business for Malaysia and, thus, has been enhancing Malaysia’s economic growth until now. Nevertheless, the industry faces many challenges, such as serious infections called basal stem rot (BSR) disease, which is caused by the fungus known as *Ganoderma boninense*. This disease has caused significant economic loss, and the total area affected by BSR in 2020 is estimated to be approximately 443,430 ha (65.6 million) oil palm trees [1]. Many measures have been implemented to control BSR diseases, such as mechanical and chemical treatments. However, the control by chemical treatment using fungicides has been shown to be less effective than the mechanical treatment and can cause harm to the environment, especially if residues remain in the soil and enter waterways [2,3]. These effects have hastened the use of biological control as a viable method to combat BSR disease in palm oil. Recently, a new fungus, *Scytalidium parasiticum*, has been isolated from the basidiomata of *G. boninense* [4], which causes BSR of oil palm in the southern region of Malaysia. The ascomycetes were detected to inhibit *G. boninense* under in vitro conditions. The potential antifungal metabolite produced by *S. parasiticum* responsible for the effects against Ganoderma remains unknown. The antifungal metabolites could be produced by *S. parasiticum* alone for the self-defense system against Ganoderma or could be induced by co-culturing both fungi; the co-culture technique has become a new trend in the search for a new biocontrol agent discovery.

The excitement in finding potential metabolites that could act as antifungal agents for the biocontrol of Ganoderma inspired us to start our experiment by co-culturing both fungi together at different time courses, from day 0 until day 5. The co-culturing strategy is inspired by nature, and studies have shown that the interactions of two or more different microbes may enhance the accumulation of constitutively present natural products [5,6] or may trigger the expression of silent biosynthetic pathways, finally yielding new compounds [7]. Furthermore, the co-cultivation approach has been widely used in the production of foods, food additives, enzymes, bulk and fine chemicals, bioremediation and the degradation of lignocelluloses. However, its application for the production of antimicrobial compounds is still in its infancy stage [8].

To achieve our objectives, our study is focused on the screening of antifungal metabolites produced by co-culturing *S. parasiticum* and *G. boninense*. The crude extracts from these interactions were applied to a preparative column to separate a consortium of active metabolites. The recycling preparative High Performance Liquid Chromatography (HPLC) is used in this study for the fractionation of extracts. This technology is used to increase the separation efficiency by recycling the sample into the column while keeping the peak dispersion to a minimum. This could be achieved by incorporating a recycle valve in the HPLC preparative system to recirculate the unresolved peaks into the column [9]. Later, the fractions were collected according to the peaks detected by a UV detector and were later tested for antimicrobial activity against *G. boninense*. The positive results of the fractions were subjected to Liquid Chromatography-Time of Flight-Mass Spectrometry (LC-TOF-MS) analysis to identify the nonvolatile secondary metabolites. As a detection technique, Ultra High Performance Liquid Chromatography-Time of Flight-Mass Spectrometry (UHPLC-TOF-MS) was preferred, since it provides enhanced sensitivity in a full-scan mode compared to triple-quadrupole MS and accurate mass measurements, which are beneficial for structure elucidation. The preprocessed data was analyzed with the appropriate statistical tools—in particular, Principal Component Analysis (PCA) [10]. The markers (metabolites) with the greatest variation in the LC-MS dataset were identified by comparing the accuracy of the *m*/*z* value and MS/MS spectra with an available database, as well as a comparison with pure standard compounds. The schematic diagram of the experimental design of this study is shown in Figure 1. The emphasis of this study is to characterize the metabolites released from the co-culture and use the metabolites that suppress *G. boninense* growth to develop a biocontrol agent against Ganoderma. The strategies devised in this project show their capability to facilitate the isolation of antifungal compounds biosynthesized by co-culturing *S. parasiticum* and *G. boninense*.

## 2. Results

### 2.1. Antifungal Activity against G. boninense

*S. parasiticum* released a yellowish pigment when co-cultured with *G. boninense*. Although the content of this yellowish exudate was unknown during this stage, the exudate has fungistatic effects that cause irregular hyphae, reduce growth and degrade the mycelia, as reported by Goh [4]. The exudates from the co-culture media were extracted, and the fractions were collected by recycling preparative HPLC for further analysis (Appendix A). The antifungal activity of the fractions from the *G. boninense* growth age of 0 day (G0), 3 days (G3) and 5 days (G5), co-cultures were observed against *G. boninense* (Table 1). The antifungal activity showed positive results in the G0 fractions, where *S. parasiticum* and *G. boninense* were co-cultured simultaneously. As shown in Table 1, the positive results of the G0 fractions were observed in the second, third and fifth recycles. The third recycle showed a higher inhibitory effect than hygromycin, the positive control used in this study. The positive results of the fractions that showed the highest inhibition zones that were analyzed by LC-TOF-MS to identify the metabolites that contribute to antifungal activity.

### 2.2. Multivariate Analysis of the G. boninense–S. parasiticum Co-culture

To evaluate differences in the acquired LC-TOF-MS base peak chromatograms, a Principal Component Analysis (PCA) was carried out on the preprocessed LC-TOF-MS data matrices. This is an initial evaluation of the developed modes and the detection of outliers that could influence their predictive ability and trends. Spectra with similar profiles were plotted closely together in the PCA (normalized Pareto scaled, log-transformed) score plot. The PCA score plot (Figure 2) showed that none of the samples were outside the Hotelling T2 95% confidence limit with an acceptable predictability of 10% (Q2X = 0.501), meaning that 50% of the total variation of the X-matrix could be predicted by the model. The X-matrix is the peak intensities generated from an extracted ion chromatogram (EIC). The first two principal components (PC1 and PC2) described the variation in the X-matrix (R2X = 0.627) with an acceptable predictability of 50% (Q2X = 0.501). The first component and second component accounted for 49% and 13% of the total variation, respectively, separating the control groups and the treatment groups (different fractions).

### 2.3. PCA Loading Plot

All variables were displayed in loading plots. The loading plots exhibited the influence (weight) of the individual X-variables in the model. Each point represents a different spectral intensity. The PCA loading plot (Figure 3) showed the potential biomarkers for distinguishing samples of all groups. Looking at the samples with positive scores in PC1 in this model, the control masses were grouped (positive score) due to the presence of 133.048 *m*/*z* at a retention time of 2.11 min and 325.106 *m*/*z* at 2.05 min, which were separated from the treatment group (Figure 3). The treatment group (negative score) was separated along PC1 due to the presence of 236.162 *m*/*z* at 2.13 min, 134.020 *m*/*z* at 2.03 min and 288.284 *m*/*z* at 9.52 min (Figure 3). The *G. boninense* growth age of 0 day that was fractioned and collected at fifth recycled with fraction number 37–40, G0 R5 37–40 and G0 R5 73–78 were separated from G0 R5 74–78 in the negative score along PC2 due to the presence of 376.256 *m*/*z* at 11.43 min in both G0 R5 37–40 and G0 R5 73–78. The G0 R5 74–78 was separated along with the PC2 positive score from the other treatment group due to the presence of 126.012 *m*/*z* at 1.59 min and a high level of 236.162 *m*/*z* at 2.13 min.

### 2.4. Variable Influence on Projection (VIP) List

The variable influence on projection (VIP) list was generated from the Partial Least Square- Discriminant Analysis (PLS-DA) model. The VIP is commonly used to summarize the importance of the X-variables in multivariate models based on projections. It summarizes the contribution a variable makes to the model. The value of the VIP score, which is greater than 1, is the typical rule for selecting relevant variables. The VIP can only be generated from a PLS-DA analysis. The most important metabolites (by mass) responsible for the apparent discrimination (those with VIP > 1) are listed in Table 2. In the component matrix, variables with higher values indicated a higher contribution of discrimination from groups of that component. From the VIP list, the extracted ion chromatogram (EIC) for each compound across 16 samples is shown in Figure 4. The EIC was extracted from profile analysis software, where the signal intensity data was preprocessed (normalized) and later uploaded in SIMCA-P (log-transformed and scaled (Pareto)).

### 2.5. Hierarchical Clustering Analysis of Metabolites

An analysis conducted by hierarchical clustering exhibited that there were several clusters (groups) present; the larger group (the right hand) was divided into three subclusters, and the left-hand group was divided into one subcluster (Control) (Figure 5A). The G0 R3 37–40 group was clustered together with the G0 R5 73–78 group. The heat map of the respective metabolites corresponding to each group was also presented (Figure 5B). A heat map is a data visualization technique that shows the magnitude of a phenomenon as color in two dimensions. The variation in color is due to the intensity of the compound based on the EIC. Rows represent metabolites, and columns represent samples.

## 3. Discussion

### 3.1. Metabolites Significantly Different in the Treatment Group and Control Group

The chemical identification of unknown metabolites remains a challenging task in metabolite profiling [11]. The identification of key metabolites (potential biomarkers) is critical for obtaining desirable discrimination results [12]. Therefore, the multivariate data analysis by using PCA and PLS-DA were employed in this study and able to provide information on the candidate biomarker from its loading plot by looking at their significant changes up and down regulation. Tryptamine or indoleamine, which was putatively identified with *m*/*z* 133.048, was found to be the most significantly different metabolite between the treatment and control groups based on the VIP value (Table 2). This metabolite was found in the control group but was absent in the treatment group (co-culture media), as the fungus had fully catabolized tryptophan and tryptamine as the sources of carbon and nitrogen (Figure 4A). Tryptamine is an alkaloid that contains an indole group and is a derivative of tryptophan [13]. On the other hand, the presence of 236.162 *m*/*z*, which was putatively identified as eudistomin I based on the Kyoto Encyclopedia of Genes and Genomes (KEGG) and MassBank database, was found at high levels, approximately 60% higher in the treatment group compared to the control group (Figure 4B).

Eudistomin I is a B-carboline alkaloid, an indole alkaloid class that is isolated from the marine Caribbean ascidian *Eudistoma olivaceum* [14]. Biosynthesis of the B-carboline alkaloid is associated with the Pictet-Spengler reaction between indoleamines (tryptamine and serotonin) [15]. Other studies showed that the synthesis of eudistomin began with tryptamine as a starting material [16]. This finding suggested that the tryptamine that was detected in the control group was used up by the co-culture fungi and involved in the biosynthetic pathway to produce eudistomin I, which explains why tryptamine was not detected in the treatment group. In addition to eudistomin I, we found another interesting metabolite in the treatment group, 296.065 *m*/*z*, which was putatively identified as penipanoid A (Figure 4C). Penipanoid A is a triazole anthranilic acid alkaloid and has been isolated from marine fungi [17]. Both penipanoid A and eudistomin I are derived from the shikimate pathway through the conversion of chorismate to anthranilate by anthranilate synthase and the degradation of tryptophan via the anthranilate branch of the β-carboline pathway for eudistomin I.

Other metabolites that were present in the control group but were not in the treatment group included cis-aconitate (175.130 *m*/*z*) (Figure 4D) and glucose (325.1 *m*/*z*) (Figure 4E). Both cis-aconitate and glucose were consumed as a source of carbon and energy for the production of aspartic acid (134.0458 *m*/*z*) (Figure 4F) and lysine (147 *m*/*z*). Lysine was among the highest metabolites found in the treatment group but was not present in the control group (Figure 4G). It is known that fungi synthesizes amino acids, including lysine, via the α-aminoadipate pathway [18]. This pathway has been assumed to be involved in the production of antifungal drugs [19]. Lysine is significant in enhancing the antifungal action of amphotericin B against *Candida albicans* strains, although lysine itself did not exert a fungicidal effect [20]. Meanwhile, aspartic acid is a precursor for other essential amino acids, such as threonine, lysine and homoserine, in a branched and complex regulated pathway [21].

Oleic acid (288.284 *m*/*z*) (Figure 4H) and stearamide (316.317 *m*/*z*) (Figure 4I) were present in the co-culture media but were absent in the control group. Interestingly, this fungal co-culture produced a high amount of fatty acids (oleic acid) compared to other metabolites, such as amino acids (please refer to the heat map, Figure 5B). Both microorganisms could be considered as oleaginous microorganisms [22]. Our findings also showed that a compound with a mass of 476.143 *m*/*z* putatively identified as naringenin 7-O-beta-D-glucoside, was produced in the treatment group and was high in the G0 R5 74–75 group compared to the other group (Figure 4K). This compound has antimicrobial effects against a wide spectrum of bacteria (Gram-positive and -negative) and acts as a single component [23].

### 3.2. Metabolites that Contribute to Antimicrobial Activity

An analysis of the heat map (Figure 5B) showed two main clusters: (a) and (b). The (a) group contained a list of metabolites that were highly abundant in the control group compared to the treatment group. It is suggested that these metabolites were utilized by both fungi in the co-culture directly from the media as carbon and nitrogen sources. In the (a) group, catechin was detected in the oil palm extraction media (OPEM), which is in agreement with a previous study that showed that oil palm leaf extract was rich in catechin [24] (Figure 4L). Interestingly, our studies provide the first evidence of catechin uptake from the media by the co-cultured fungi. Other metabolites that were taken up by the fungi were mainly carbon and nitrogen sources, such as glucose, tryptamine, tryptophan and aconitate.

The (b) group contained a list of metabolites that were highly abundant in the treatment group, and presumably, these metabolites were secreted by the treatment group. Tyrosine, eudistomin, naringenin and penipanoid A (metabolites in subcluster (c)) were in high abundance in the G0 R5 74–78 group compared to the other treatment group. The antifungal activity of this group was not significantly different from the other treatment groups, suggesting that these metabolites contributed less to the antifungal activity. The contribution of these metabolites to antifungal activity, however, cannot be ignored. It has been reported that the eudistomin group, which contains an oxathiazapine ring, exhibits a wide antibacterial spectrum, including activity against *Enterobacter cloacae*, *Escherichia coli*, *Klebsiella pneumoniae*, *Proteus vulgaris*, *Pseudomonas aeruginosa*, *Salmonella typhimurium*, *Serratia marcescens*, *Staphylococcus aureus* and *Staphylococcus epidermidis* in disc diffusion assays [25]. Nevertheless, a previous report showed that penipanoid A did not exhibit antibacterial activity against two bacteria (*Staphylococcus aureus* and *Escherichia coli*) or antifungal activity against five plant-pathogenic fungi (*Alternaria brassicae*, *Fusarium oxysporium* f. sp. *vasinfectum*, *Coniella diplodiella*, *Physalospora piricola* and *Aspergillus niger*) [17].

In addition, we found an interesting metabolite from the treatment group, which is putatively identified as homoserine lactone (Figure 4J and Figure 5B). This compound acts as a quorum-sensing molecule [26]. Generally, fungi utilize homoserine lactone to maintain or alter their population’s behavior, including changes in the morphological expression and reproduction, such as sporulation, spore dormancy and germination. Apart from their signaling ability, homoserine lactones trigger the production of other metabolites in the fermentation process. To date, it is undeniable that these molecules play an important role in triggering the production of active fungistatic metabolites, such as penipanoid A and eudistomin I, as defense mechanisms. For example, the treatment with N-butyryl-dl-homoserine lactone resulted in the production of di- and trisulfide emestrins A in *Trypanasoma brucei* and *Leishmania donovani* [27]. For the next biocontrol screening experiment, homoserine lactones could be added to the co-culture media to increase the production of metabolites with antimicrobial properties.

On the other hand, the inhibition and interruption of quorum-sensing molecules is also a promising strategy to combat pathogen infection [28]. This could be done to combat Ganoderma growth by the inhibition or degradation of its quorum-sensing molecules. Specifically, triggering or inhibiting the production of quorum-sensing molecules has a huge effect on both biocontrol strategies in combating a Ganoderma infection.

Metabolites that are in subcluster (d) (Figure 5B) enhanced the antimicrobial activity where the abundance of these metabolites was consistent with the increased antimicrobial activity between treatment groups, G0R3 37–41. Oleic acid and stearamide are both long-chain unsaturated fatty acids that are well-documented for their antimicrobial properties, especially in food additives [29,30]. In fact, fatty acid biosynthesis is an emerging target for the development of antimicrobial therapeutic drugs. To date, there are no studies on the antimicrobial effects of oleic acid against pathogenic *Ganoderma* sp. fungi. However, studies by Walters et al. [31] demonstrated that oleic acid completely inhibited the growth of plant pathogenic fungi *(Rhizoctonia solani*, *Pythium ultimum*, *Pyrenophora avenae* and *Crinipellis perniciosa*) at 1000 µM. The groups G0 R5 74–78 and G0 R5 73–78 showed antimicrobial activities with inhibition zones of 15 mm and 17 mm, respectively, and were as effective as the positive controls and G0R3 37–41 (Table 1). The G0 R5 73–78 and G0R3 37–41 groups showed higher inhibition zones compared to G0 R3 74–78 due to the high contents of oleic acid, stearamide and tricasonoyl ethanolamide.

Despite these metabolites being clustered, clusters c and d might work synergistically to exhibit antimicrobial effects. Many studies have been reporting synergy between combinations of flavonoids that showed stronger antimicrobial activity than the single compound [32]. Our studies involving the search for fungistatic metabolites have become known, as we are able to identify a few metabolites that exhibit antimicrobial activity. Previous studies performed by Goh et al., 2016 [33] hypothesized that the yellowish exudates released from *S. parasiticum* killed and inhibited the growth of *G. boninense*. Assuming that the metabolites from yellowish exudates that are involved in antimicrobial activities are not accurate and valid, as it has been clearly stated that some metabolites must work synergistically to have antimicrobial effects, for future experiments, we would like to isolate a single compound from this co-culture media—particularly, the yellowish compound—to prove this hypothesis. The detailed profiling result of group G0 R3 37–41 (the largest inhibition zone) is included in Table 3 and with its chromatogram in Figure 6.

### 3.3. Metabolites Involved in the Biosynthetic Pathway of the G. boninense–S. parasiticum Interaction

All the metabolites involved in the *G. boninense–S. parasiticum* interaction were mapped to biological pathways in the KEGG database; these metabolites and pathways involved alkaloids from the tryptophan biosynthetic pathway, phenylalanine biosynthetic pathway, flavonoid biosynthetic pathway and the fatty acid biosynthesis pathway (Figure 7). The results indicated that the fungus–fungus interactions led to many metabolite alterations and the production of secondary metabolites as defense mechanism against other microorganisms.

## 4. Materials and Methods

### 4.1. Study Design

*S. parasiticum* was grown in the *G. boninense* broth culture at different times, measured in accordance to the *G. boninense* growth age (0 day, 3 days and 5 days). After 14 days, the co-cultures were extracted with 60% methanol. The extract was concentrated and fractionated with recycling preparative HPLC. The fractions were tested for antifungal activity against *G. boninense*. The fractions that showed the highest inhibition zones were sent to LC-TOF-MS for further analysis. The detailed workflow of the experimental design is shown in Figure 1.

### 4.2. Fungal Strains

The fungal strains were obtained from Advanced Agriecological Research Sdn Bhd. *G. boninense* G10 (isolated from the Batu Lintang oil palm estate located in Kedah, Malaysia) [34] and *S. parasiticum* AAX0113 (isolated and identified from the *G. boninense* culture collected from the Fraser Estate, located in Johor State, Malaysia) [4] and were used in this study.

### 4.3. Chemicals

Methanol and acetonitrile were purchased from Merck (Darmstadt, Germany). Formic acid for UHPLC-TOF-MS analyses were purchased from Sigma-Aldrich (Steinheim, Germany). Water was purified using a Milli-Q system (Millipore, Bedford, MA, USA). All solvents used were of HPLC grade.

### 4.4. Extraction and Isolation

#### 4.4.1. Preparation of Oil Palm Extract Broth (OPEB)

Healthy oil palm trunks were collected from the oil palm estate and processed without agar [4].

#### 4.4.2. Experimental Design of Culture Conditions

*G. boninense* G10 and *S. parasiticum* AAX0113 fresh cultures were prepared and maintained on malt extract agar (MEA) (Difco, Becton Dickinson, Franklin Lakes, NJ, USA) at 24 °C in dark conditions for 7 days before inoculating them onto the OPEB. Five 10-mm agar plugs of *G. boninense* were precultured in the 250–mL conical flask and incubated at 24 °C on an orbital shaker at 120 rpm. Then, five 10-mm agar plugs of *S. parasiticum* were transferred onto the same conical flask with the different time of *G. boninense* growth age, (0, 3 and 5 days) which was preinoculated in the media. The co-culture flasks were then labeled as G0, G3 and G5 according to the *G. boninense* growth age. Each flask continued to be incubated up to 14 days. Exudates produced during the *G. boninense–S. parasiticum* co-culturing assays were collected after the indicated incubation period and stored at −80 °C before extracting using 60% cold methanol. The control flasks with modified OPEM were without *S. parasiticum* inoculum or mycelial plugs. All treatments with or without *S. parasiticum* were performed in three to five replications to produce enough exudates for metabolite extraction.

#### 4.4.3. Extraction of the Compound

*S. parasiticum* and *G. boninense* were cultivated on oil palm extraction media (OPEM) and stored at −80 °C until extraction. The extraction method was optimized from Lim [35]. Approximately 50 mL of cold 60% methanol was added to the sample at a ratio of 1:1 (i.e., 50 mL of sample in 50 mL of cold 60% methanol). After a vigorous mix (vortex ~30 s), the samples were subjected to sonication in an ice bath with a sonic dis-membrator FB120 (Fisher Scientific, Waltham, MA, USA) fitted with a Model CL-18 probe at 65% power and 30% amplitude with 15-s pulses. The sample was then centrifuged at 10,000 rpm for 10 min at 4 °C. The methanol extracts were stored at −80 °C before the next isolation steps, which were performed with an HPLC preparative system.

#### 4.4.4. Recycling Preparative HPLC Fractionation of the Sample

The recycling preparative HPLC system consisted of an 880-PU pump (Jasco, Tokyo, Japan), a manual injector (Model 7125, Rheodyne, Cotati, CA, USA) with a 20.0-mL sample loop and an 875-UV variable wavelength detector. The detector was set at 280 nm, 254 nm and 320 nm, respectively. The preparative column was a JAIGEL C18 reversed-phase column (250 × 30 mm, particle size 10 µm; Osaka, Japan). The mobile phase was in isocratic mode, 98% methanol. The flow rate was 10 mL/min, and the injection volume was 10 mL. The fraction highlighted in Appendix A was collected and dried in a rotary evaporator. The dried fraction was tested against a Ganoderma culture, and the inhibitory activity was measured.

### 4.5. Anti-Ganoderma Activity in the Laboratory

Antifungal activities of the dried fraction were first tested against Ganoderma cultures. Each fraction was diluted in distilled water [36]. *S. parasiticum* mycelial plugs (10-mm diameter, from a 7-day-old culture) were then excised and transferred to potato dextrose agar (PDA) plates; hygromycin was used as a positive control. The plates were then incubated at 24 °C and observed after 24 h. The inhibition zone on the agar media was defined by the diameter of the clear zone or a zone without fungal growth. The fraction that showed positive inhibition against Ganoderma was sent to the LC-TOF-MS analysis to identify the active compound(s).

### 4.6. LC-TOF-MS Analysis

Analyses were performed using ultra-high-performance liquid chromatography (UHPLC) with a microTOF Q III mass spectrometer (MS) (Bruker Daltonics, Bremen, Germany) equipped with an electrospray source (ESI) and connected to an Ultimate 3000 UHPLC system (Dionex, Sunnyvale, CA, USA) equipped with an Acclaim™ Polar Advantage II, 3 × 150 mm, 3-µm particle size C18, reverse-phase column. The gradient elution was performed at 0.4 mL/min at 40 °C using water with 0.1% formic acid (A) and 100% acetonitrile (B) with a 22-min total run time. The sample injection volume was 1 µl. The gradient was as follows: 5% B (0–3 min), 80% B (3–10 min), 80% B (10–15 min) and 5% B (15–22 min). MS was performed in ESI positive ionization mode with the following settings: capillary voltage, 4500 V, nebulizer pressure, 1.2 bar and drying gas, 8 L/min at 200°. The scan range was from 100–1000 *m*/*z*. Data processing was performed using the software Data Analysis 4.0 and Profile Analysis (Bruker Daltonics) [37].

### 4.7. Data Processing and Data Analysis

All mass spectral data were acquired using Data Analysis software (version 4.0, Bruker Daltonics). Raw data (d) files were imported into Profile Analysis software (Bruker Daltonics). The Profile Analysis software was used for further data processing, including peak alignment and peak normalization by using a special algorithm generated from an extracted ion chromatogram (EIC) [38]. The parameters used were Retention Time (RT) range 0–20 min, mass range 100–1000 Da and mass tolerance 0.02 Da. Internal standard detection parameters were excluded for the peak retention time alignment. Furthermore, isotopic peaks were excluded from the analysis. The noise elimination level was set at 10.00, the maximum masses per RT was set at 6 and, finally, the RT tolerance was set at 0.01 min. Multivariate statistical evaluation of the preprocessed metabolic profiling data was performed with SIMCA-P + (version 12) (Umetrics, Umeå, Sweden). We performed the pretreatment method used in metabolomics data where the signal intensity data were log-transformed and scaled (Pareto) [39] and, finally, a partial least squares (PLS) model was generated, which was applied to selected statistically significant variables that were able to discriminate the compared groups. All multivariate models were built on previously prepared data with pareto scaling and logarithmic transformation [40].

Other data analyses, such as a Hierarchical Analysis (HCA) and heat map, were performed by using MetaboAnalyst 3.0 software [41] to visualize the metabolite profiles and reveal the relationship between metabolites and samples. The analysis was performed by using the extracted dataset by 17 metabolites filtered by ANOVA, *p* < 0.005.

### 4.8. Metabolite Identification

Compound identification of metabolites was performed by comparing of accuracy of the *m*/*z* value (<20 ppm, and MS/MS spectra with an in-house database: Human Metabolome Database (HMDB), Metabolite and Chemical Entity Database (METLIN) and KEGG. Some of the metabolites were compared with a commercially available reference standard (retention time 0.01 min and mass accuracy less than 3 ppm, MS/MS spectra).

### 4.9. Statistical Analysis

The differential metabolites obtained from the multivariate data analysis software were then validated using ANOVA with post-hoc Tukey’s tests. All the metabolites that contributed to group separation were significant at *p* < 0.005.

## 5. Conclusions

Our approach of co-culturing two different microorganisms to determine the biocontrol agents of Ganoderma has shed new light. Current trends for identifying new antimicrobial compounds include co-cultivating two or more microorganisms together. The interactions may trigger the formation of new secondary metabolites that may not produce in a single culture system. The results presented here showed that potential antimicrobial metabolites (alkaloids, flavonoids and fatty acids) were triggered when both *G. boninense* and *S. parasiticum* were cultured together. This finding suggested that a co-culturing technique for biocontrol determination should be introduced at the initial stage of Ganoderma manifestation in order to ensure its effectiveness as inhibitors. Although the anti-Ganoderma activity showed positive results in laboratory trials, we believed that, by identifying the metabolites released during the fungus interaction, it will lead us to the more effective control of Ganoderma in the future. This technique can be integrated into the practical management of Ganoderma control in an oil palm field by focusing on the production of the metabolites, minimum and complete inhibition through the optimal concentration of metabolite extracts and its capacity to remove Ganoderma completely. When oil palms are free from Ganoderma infection, it contributes to a higher yield and productivity in the palm oil industry that benefits Malaysia and other palm oil-importing nations.

## Figures and Tables

**Figure 1 molecules-25-05965-f001:**
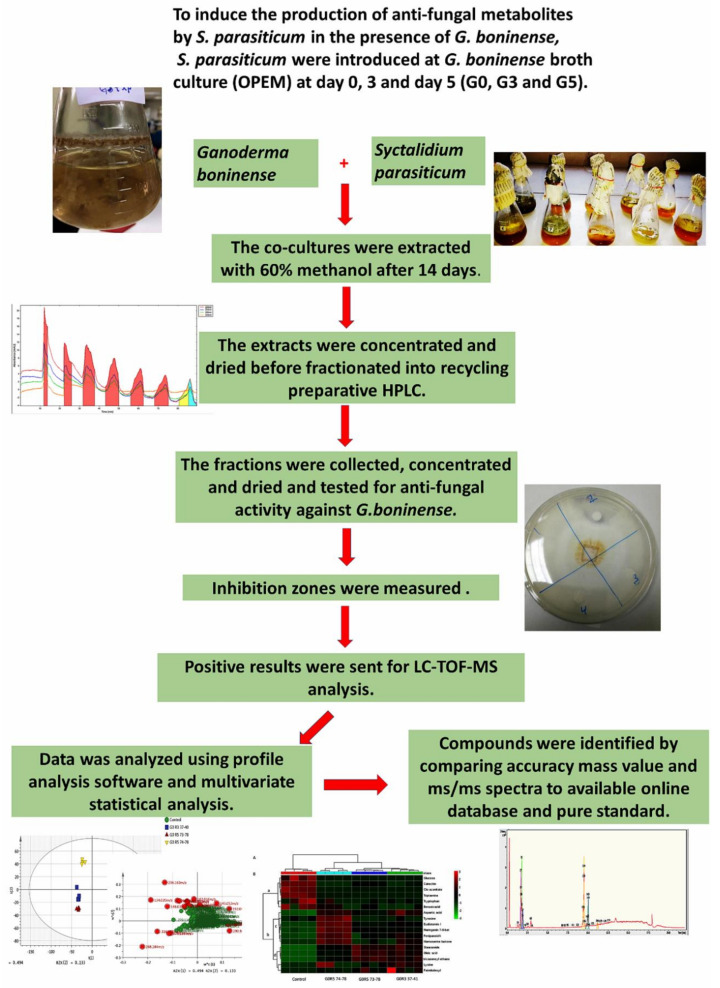
Schematic diagram of the study. The method consists of co-culturing both fungi, *Ganoderma boninense and Scytalidium Parasiticum*, in the same media with different growth ages (0, 3 and 5 days); liquid extraction; fractionation into recycling preparative High Performance Liquid Chromatography (HPLC); antifungal activity, a mass spectrometry analysis, including a data analysis and compounds identification.

**Figure 2 molecules-25-05965-f002:**
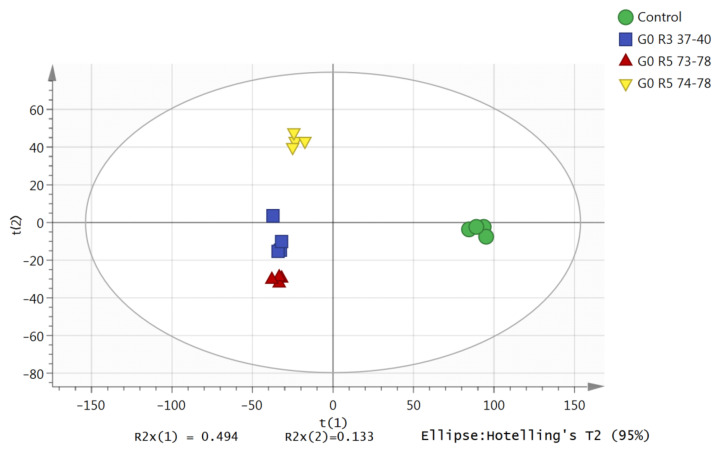
Principal Component Analysis (PCA) score plot of the metabolic changes of the control and co-culture of the *S. parasiticum* and *G. boninense* fractions in the first two principal components (PC1 and PC2). G0, G3 and G5 refer to the *G. boninense* growth age (0, 3 and 5 days), respectively. R3 and R5 represent fractions recycled number at number 3 and number 5, respectively.

**Figure 3 molecules-25-05965-f003:**
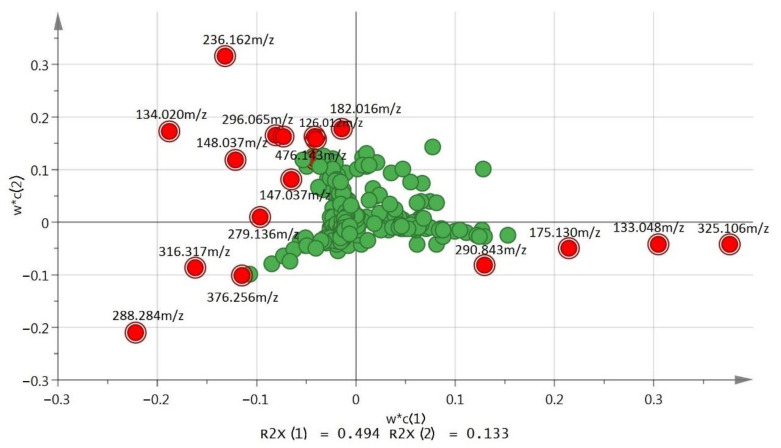
PCA loading plot of control and treated samples that showed the distribution of metabolites (masses) in w*c(1) and w*c(2) planes (green dots). The most important metabolites in VIP list are highlighted in red bulleted dot.

**Figure 4 molecules-25-05965-f004:**
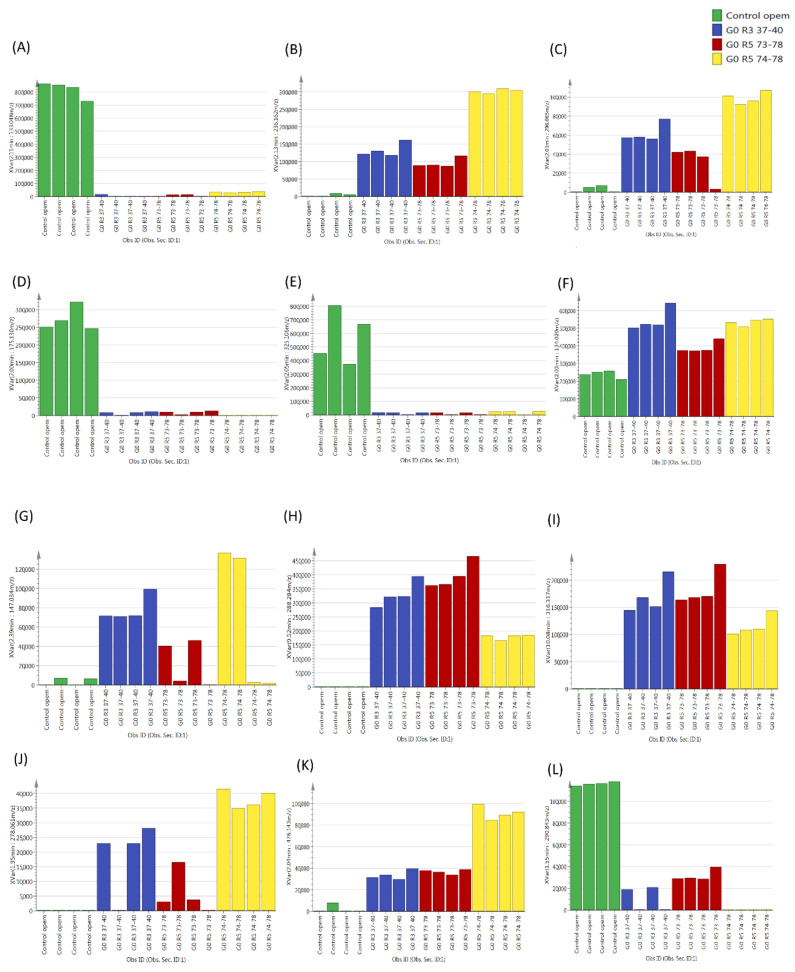
Representative ion intensity for the *m*/*z* value (**A**) 133.048 (Retention Time (RT) 2.11 min), (**B**) 236.162 (RT 2.13 min), (**C**) 296.065 (RT 2.01 min), (**D**) 175.130 (RT 2.00 min), (**E**) 325.106 (RT 2.05 min), (**F**) 134.020 (RT 2.03 min), (**G**) 147.034 (RT 2.39 min), (**H**) 288.284 (RT 9.52 min), (**I**) 316.317 (RT 10.08 min), (**J**) 278.061 (RT 1.95 min), (**K**) 476.143 (RT 2.04 min) and (**L**) 290.843 (RT 1.55 min) across 16 samples.

**Figure 5 molecules-25-05965-f005:**
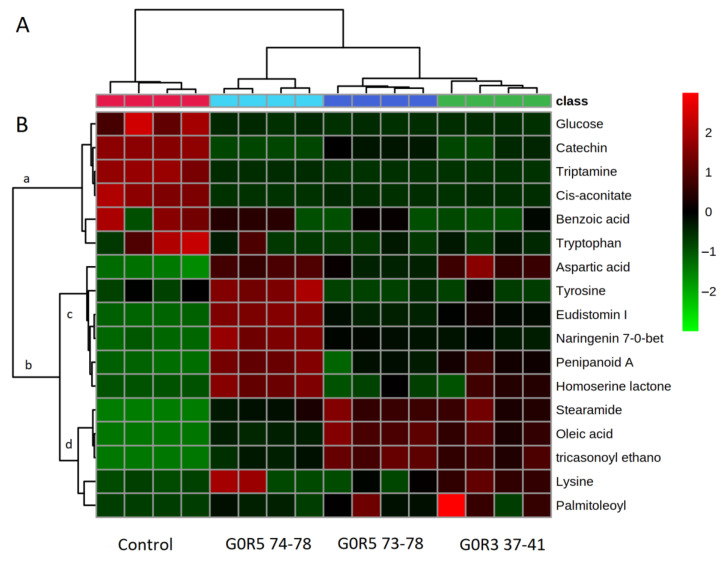
Dendrogram obtained after the hierarchical classification of metabolite profiles in the control and *G. boninense–S. parasiticum* co-culture fractions. (**A**) Heat map of metabolites are shown. (**B**) Bright red denotes highest intensities of metabolites, and light green denotes the lowest intensities or complete absence of metabolites.

**Figure 6 molecules-25-05965-f006:**
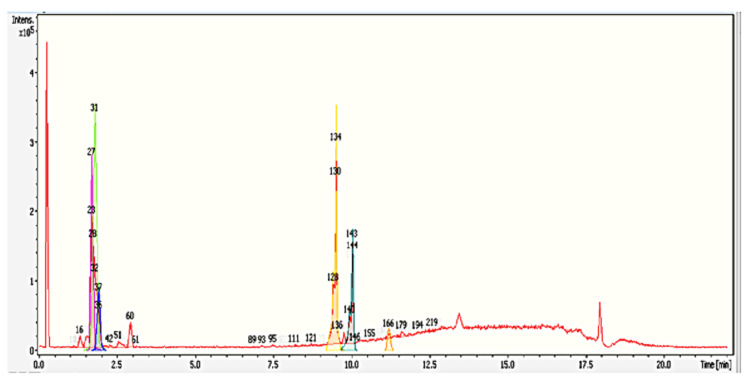
Base peak chromatogram (BPC) (red color) and dissect chromatograms (different colors) of the G0R3 37–41 fraction by Liquid Chromatography-Time of Flight-Mass Spectrometry (LC-MS-TOF) in a positive ionization mode. Peak labeling with different numbers represent the compounds identified that correspond to Table 3. Bruker’s dissect algorithm in Data Analysis software 4.0 allows the user to observe overlapping peaks at very similar retention times.

**Figure 7 molecules-25-05965-f007:**
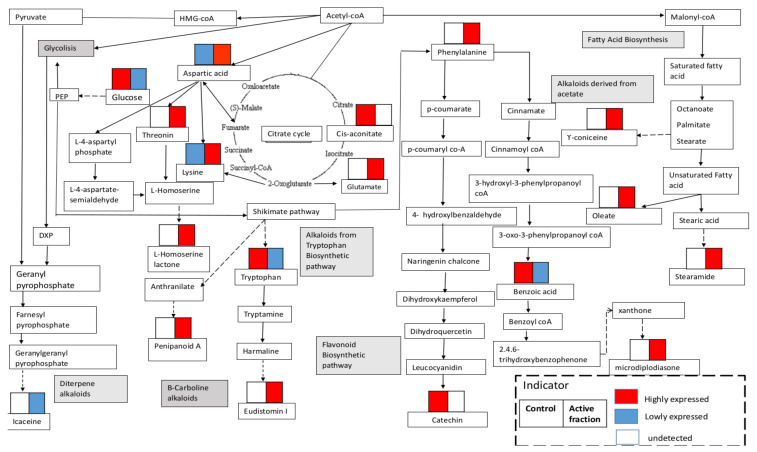
Metabolites involved in the biosynthetic pathway of *G. boninense–S. parasiticum* interactions. Single arrows represent one-step enzymatic conversions, while dashed arrows represent multiple reactions.

**Table 1 molecules-25-05965-t001:** Growth inhibition zone in mm indicates the antifungal activities of recycling preparative High Performance Liquid Chromatography (HPLC) fractions against *Ganoderma boninense.*

Sample	No. of Recycle	Retention Time (min)	Test Concentration (µg/mL)	Diameter (mm)
Hygromycin (Positive control)	-	-	100	17
Distilled water (negative control)	-	-		-
XP with G0	3rd	37–41	100	20
XP with G0	2nd	37–40	100	8
XP with G0	5th	73–78	100	17
XP with G0	5th	74–78	200	15
XP with G0	6th	72–85	800	-
XP with G0	1st	24–28	500	-
XP with G3	3rd	49–51	300	-
XP with G3	4th	56–61	600	-
XP with G3	5th	55–92	500	-
XP with G3	7th	105–107	50	-
XP with G3	8th	112–120	700	-
XP with G5	3rd	50–53	400	-
XP with G5	6th	87–89	200	-
XP with G5	8th	111–119	500	-
XP with G5	9th	124–129	700	-

XP refers to *Scytalidium parasiticum;* G0, G3 and G5 refer to the *G. boninense* growth age (0, 3 and 5 days), respectively. XP with G0 refer to *S. parasiticum* and *G. boninense* that were cultured simultaneously in the same media. XP with G3 and G5 refer to *S. parasiticum* was cultured in the same media after 3 and 5 days of *G. boninense* growth, respectively.

**Table 2 molecules-25-05965-t002:** Putatively identified metabolites with the highest score of the Variable Influence on Projection (VIP) as determined by a Partial Least Square-Discriminant Analysis (PLS-DA).

Var ID (Primary).	M3.VIP (2) 2.44693 *	M3.VIP (2) cvSE	Name of Metabolite
2.11 min: 133.048 *m/z*	7.8737	1.5332	Tryptamine
2.13 min: 236.162 *m/z*	7.07568	2.49579	Eudistomin I
2.05 min: 325.106 *m/z*	6.40082	2.65825	Glucose
9.52 min: 288.284 *m/z*	6.35539	1.37838	Oleic acid
2.03 min: 134.020 *m/z*	5.28688	1.82764	Aspartic acid
2.00 min: 175.130 *m/z*	4.58414	0.974071	Cis-aconitate
2.01 min: 296.065 *m/z*	3.82321	0.662272	Penipanoid A
2.04 min: 476.143 *m/z*	3.70863	2.01345	Naringenin 7-O-beta-D-glucoside
2.26 min: 182.016 *m/z*	3.70402	1.55685	Tyrosine
1.55 min: 290.843 *m/z*	3.16889	0.535434	Catechin
1.95 min: 278.061 *m/z*	2.60836	0.638658	C10-Homoserine lactone
2.39 min: 147.034 *m/z*	2.15276	3.25775	Lysine

M3. VIP (2) 2.44693 * refers to PLS-DA model which generate VIP value that has been coefficiently combined and plotted in a w*c(1) and w*c(2); cvSE refers to Jack-knife standard error of the VIP computed from all rounds of cross validation in SIMCA-P analysis.

**Table 3 molecules-25-05965-t003:** Metabolites that were detected in G0 R3 37-41.

Peak No	RT	Mass per Charge Ratio *m*/*z* Measured Mass	Assigned Identity	Collision Energy	Molecule Formula	MS/MS Fragmentation (Intensity)	Adduct
10	1.1	110.0108	Hypotaurine *	15.5eV	C_2_H_7_NO_2_S	92.0250, 1143	(M + H)
15	1.3	122.9258	Benzoic acid *	16.1eV	C_7_H_6_O_2_	105.08760, 1290	(M + H)
16	1.3	290.8457	Catechin	24.5eV	C_15_H_14_O_6_	171.5677, 3678 171.5887, 4058 171.6049, 4058 179.1129, 3312 179.1367, 4082 179.1813, 4082	(M + H)
17	1.3	275.2777	5-Methyl-2-thiouridine	23.7eV	C_10_H_14_N_2_O_5_S	163.1166, 3705 163.1334, 3914 163.1570, 3914	(M + H)
20	1.5	202.4605	Unknown	20.1eV		130.6173, 4463 130.6329, 4463 130.6502, 4463 202.1813, 9614	
24	1.7	130.0509	1-pyrroline-3-hydroxy-5-carboxylic acid	16.5eV	C_5_H_7_NO_3_		(M + H)
26	1.7	120.0667	Threonine *	16.0eV	C_4_H_9_NO_3_	119.0912, 3282	
27	1.7	147.0778	Lysine *	17.4eV	C_6_H_14_N_2_O_2_	130.0551, 18195	(M + H)
31	1.8	236.1492	Eudistomin I	21.8eV	C_17_H_17_N	131.0746, 2460 144.1066, 1842 159.0711, 1092 162.1228, 1025 166.1661, 1081 235.9925, 1923 236.1602, 468519	(M + H)
32	1.8	134.0458	Aspartic acid *	16.7eV	C_4_H_7_NO_4_	130.6177, 4202 130.6332, 4202 130.6488, 4202	(M + H)
34	1.8	476.1605	Naringenin 7-O-beta-D-glucoside	33.8eV	C_22_H_25_N_3_O_7_S	189.0968, 4684 189.1137, 4684 423.7170, 3421 442.7981, 3304 442.8188, 3304 442.8545, 3304 442.8990, 3304	(M + CAN + H)
35	1.8	296.065	Penipanoid A	24.8	C_16_H_13_N_3_O_3_	134.0476, 7620 146.0429, 7772 176.6487, 4717 212.0545, 9702 232.0834, 7284 260.0760, 31695 278.0874, 15708	(M + H)
36	1.9	279.0883	Microdiplodiasone	23.9eV	C_14_H_14_O_6_	154.1383, 3673 157.9724, 4158 157.9956, 4158 158.0143, 4158 165.1407, 4022 165.1613, 4022 169.1546, 4371	(M + H)
37	1.9	278.0867	C-10 Homoserine lactone	23.9eV	C_15_H_14_N_4_O_4_	154.1383, 3673 157.9724, 4158 157.9956, 4158 158.0143, 4158 165.1407, 4022 165.1613, 4022 169.1546, 4371	(M + Na)
41	2.23	148.037	Glutamic acid *	17.5	C_5_H_9_NO_4_	130.0593, 9726	(M + H)
42	2.26	182.016	Tyrosine *	18.8	C_9_H_11_NO_3_	136.0866, 551563 123.0543, 246978	(M + H)
58	2.9	152.0994	Phenylglycine *	17.6eV	C_8_H_9_NO_2_	151.8762, 2975	(M + H)
59	2.9	173.7029	unknown	18.7eV		111.0084, 107 129.0178, 268 173.0086, 87	(M + H)
60	2.9	166.1228	Phenylalanine *	18.4	C_9_H_11_NO_2_	120.079, 10000 121.082, 840 122.085, 1025 131.048, 1105 149.057, 350	(M + H)
132	9.5	288.2897	Oleic acid	24.4eV	C_18_H_34_O_2_	244.2584, 4646 270.2787, 36181 271.2816, 6490 275.6056, 8070 276.5666, 4324 288.2896, 622641	(M + H)
138	9.8	304.2834	Palmitoleoyl	25.2eV	C_16_H_29_O	145.9576, 2858 145.9740, 2858 256.2629, 14828 291.7380, 2785	(M + Li)^+^
141	10.0	316.3202	Stearamide	25.8eV	C_18_H_37_NO	298.3091, 6149	(M + CH3OH + H)
166	11.2	158.1532	Gamma-coniceine	17.9eV	C_8_H_15_N	138.1894, 2298 142.0622, 2510 145.9096, 2543 145.9323, 2970 145.9569, 2970 145.9714, 2970	(M + CH3OH + H)
175	11.6	398.2401	Tricasonoyl ethanolamide	29.9eV	C_22_H_37_NO_2_	398.2399, 128383	(M + H)
176	11.6	376.2576	Icaceine	28.8eV	C_22_H_33_NO_4_	292.2000, 4970 293.1886, 5565 298.6966, 7171 298.7256, 7171 302.1885, 7183	(M + H)

* Compounds that were identified by the authentic standard.

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
