# Peer review of "Metabolic Profile of Scytalidium parasiticum-Ganoderma boninense Co-Cultures Revealed the Alkaloids, Flavonoids and Fatty Acids that Contribute to Anti-Ganoderma Activity"

_molecules, 2020, doi:10.3390/molecules25245965_

Round 1

Reviewer 1 Report

The study by Ahmad et al., suggests that the metabolites such as eudistomin 1, naringenin 7-o-beta D-glucoside and penipanoid A, produced by Syctalidium parasiticum act as antimicrobial agents against a pathogenic fungus Genoderma boninense. Although this study is interesting and suggests that Syctalidium parasiticum can be used as a biological control in treating Palm tree fungal disease, the manuscript can be improved further by precise writing and organizing the ideas more carefully. Some comments to improve the study are given below,

  1. On the basis of the evidences provided in the study, the authors cannot conclude the following statement “Interestingly, we found that eudistomin I, naringenin 7-O-beta-D glucoside and penipanoid A, which were present in all active fractions, were the metabolites that contributed to the antimicrobial activity”. Instead this study suggests that the compounds showing different abundance in control and treatments could be the antimicrobial metabolites.
  2. Can authors please describe the data shown in Figure 7 in the result section and then discuss the results obtained later in the discussion section. Figure 7 is really hard to read and understand, please re do the figure. The figure legend needs work too, as the authors need to describe what are a, b, c.. etc are. They have explained in the text, but figures need to be self-explanatory.
  3.  
  4. The authors need to describe G0, G3, and G5 (they are the fractions collected from HPLC, but it needs to be stated clearly both in text body and the figure legend). The figure 1 will be more informative and easy to understand if authors indicate which fraction is G0, G3, and G5
  5.  
  6. The table legend describes "Growth inhibition Zone in cm" and the table describes "growth diameter in (mm)? 
  7.  
  8. I assume xp is S.parasiticum? Please describe all the acronym used in this study.
  9.  
  10. Figure 2: hard to see the difference between negative and positive control. This figure needs improvement.
  11.  
  12. It will be great to spell out the OPEM in figure legend as well. Considering that the treatments were also performed in OPEM then authors might not need to indicate the medium specifically for the control. 
  13.  
  14. Lines 257-259: The sentence is hard to understand.

Author Response

Response to Reviewer 1 Comments

1.On the basis of the evidences provided in the study, the authors cannot conclude the following statement “Interestingly, we found that eudistomin I, naringenin 7-O-beta-D glucoside and penipanoid A, which were present in all active fractions, were the metabolites that contributed to the antimicrobial activity”. Instead this study suggests that the compounds showing different abundance in control and treatments could be the antimicrobial metabolites.

Response 1: We have made the changes according to reviewer suggestion in  the abstract. Please refer to this sentences "Interestingly, we found that eudistomin I, naringenin 7-O-beta-D-glucoside and penipanoid A, which were present in different abundance in all active fractions except in control could be the antimicrobial metabolites"

2. Can authors please describe the data shown in Figure 7 in the result section and then discuss the results obtained later in the discussion section. Figure 7 is really hard to read and understand, please re do the figure. The figure legend needs work too, as the authors need to describe what are a, b, c.. etc are. They have explained in the text, but figures need to be self-explanatory.

Response 2:The figure title has been corrected accordingly with a new explanatory title. We add separate files with high resolution figure so that the data is well presented and clear. We make some amendment in numbering the figure. Figure 7 has been changed to Figure 3.  Figure 1 (chromatogram of recycling preparative HPLC) has been moved to supplementary figure (S1). Figure 2 (antimicrobial plate), Figure 4 (PLS-DA score plot) and Figure 5 (PLS-DA scatter plot) has been discarded from the text as suggested by another reviewer to exclude PLS-DA data as PCA has shown good separation for the treatment and control. 

3. The authors need to describe G0, G3, and G5 (they are the fractions collected from HPLC, but it needs to be stated clearly both in text body and the figure legend). The figure 1 will be more informative and easy to understand if authors indicate which fraction is G0, G3, and G5. The tested sample in figure 1 comes from G0 fraction, where S.parasiticum and G.boninense  

Response 3: The G0, G3 and G5 have been explained in Figure 7 (schematic diagram of the study), section 4.4.2 experimental design of culture conditions and in the footnote in Table 1. G0, G3 and G5 were samples from co-culture and have been extracted and fractionated in recycling preparative HPLC. The tested sample in figure 1 (now figure S1) is an example of chromatogram of recycling preparative HPLC where the highlighted zone is the fraction that were collected for antimicrobial test. Figure 1 has been moved to Supplementary data as suggested by another reviewer.

4. The table legend describes "Growth inhibition Zone in cm" and the table describes "growth diameter in (mm)? 

Response 4: It should be a “Growth inhibition zone in mm unit. We have made the correction. Please refer to Table 1 title.  

5.I assume xp is S.parasiticum? Please describe all the acronym used in this study.

Response 5: Yes XP is S.parasiticum. We have explain the acronym for S.parasticium in Table 1 footnote for a clarity.

6. Figure 2: hard to see the difference between negative and positive control. This figure needs improvement.

Response 6 : The figure is merely an additional information. Hence, we have decided to remove it from the text. The actual data of inhibition zone was shown in Table 1.

7.It will be great to spell out the OPEM in figure legend as well. Considering that the treatments were also performed in OPEM then authors might not need to indicate the medium specifically for the control. 

Response 7: OPEM is the acronym for oil palm extraction media. The word ‘control OPEM’ use in the table and figure legend was change to ’control’ as suggested by the reviewer. We have spell out the OPEM in section 4.3 Extraction of the compound.

8.Lines 257-259: The sentence is hard to understand.

Response 8: We apologized for the missing words that makes the sentences hard to understand. We have rephrase the sentence "For the next biocontrol screening experiment, homoserine lactones could be added to the co-culture media to increase the production of metabolites with anti-microbial properties" (Refer to 3.2. Metabolites that contribute to antimicrobial activity)

Reviewer 2 Report

“Metabolic profile of Scytalidium parasiticum-Ganoderma boninense co-cultures revealed the alkaloids, flavonoids and fatty acids that contribute to anti-Ganoderma activity” by Rafidah Ahmad et al., presents a study of the interactions of Scytalidium parasiticum and Ganoderma boninense, two fungi involved in basal stem rot disease of the oil palm trees. Based on previously published data it was concluded that Scytalidium parasiticum is capable of inhibiting a growth of Ganoderma boninense species, so it has become an attractive target as a new biocontrol agent. Rafidah Ahmad et al. have used a G. boninense and S. parasiticum co-cultivation approach to identify small molecules (metabolites) involved in fungi communication and specifically in regulation of the growth of the parasitic species.

I think that experimental design of this study and selected methods have a great potential to reveal valuable details of G. boninense and S. parasiticum interactions. Regrettably, the manuscript is poorly written. There are very few typos, so it looks like the effort was taken to prepare manuscript for submission. However, the text is full of grammatical errors: unclear sentences, inappropriately used words, two-sentence paragraphs (!) and future tense used to describe the results (!!). The only acceptable parts of this manuscript are the Abstract and Figures (except Fig7 which is illegible). Sadly, all figures are missing their figure legends. I found the Abstract interesting, but the Introduction is a bit disappointing since it inadequately sets the stage for the presented research. Unfortunately, the Results section contains information which would be more fitting for the figure legend/s and the Discussion contains mostly observations. In addition, the excessive use of the colloquial language sometimes gives the reader the impression of a lack basic understanding of applied statistical methods or identification tools.

Even though the manuscript requires significant revisions this is not the critical issue. It looks like the entire analysis has been completed based on peak intensities from base peak chromatograms (BPC). This is a serious mistake, since the BPC represents only the intensity of the most intense peak at every point in the analysis; therefore, it has very little analytical value. Based on that I cannot consider this manuscript for publication. Below you can find a list of major/minor issues as well as the list of specific examples.

Other major issues to address before submission:

  1. I think that an “untargeted screening approach” is a little bit of the overstatement. The extraction method imposes a bias to what kind of molecules are extracted from the co-culture. Further, additional bias is introduced by using 98% methanol to fractionate extracts. It is not a surprise that alkaloids, flavonoids and fatty acids were among the identified metabolites.
  2. It is unclear how the authors determined the relative concentration of identified compounds (Table3).
  3. It is unclear how metabolites were identified. There seems to be some confusion regarding “putative” and “confident” identification. In addition, I find it hard to believe that the essential amino acids had to be identified using a standard only because there was no fragmentation data.

Minor issues to address before submission:

Lines 172-173: “unknown metabolites” means that they cannot be found in any data base. Therefore, the following statement is not true: “In this study, possible metabolites that were expressed in response to G. boninense-S. parasiticum co-culture were reported for the first time.”

Lines 207-215: Irrelevant information to manuscript scope: “Tanimura reported that microorganisms with more than 20% fatty acids as their dry weights could be considered oleaginous microorganisms [26]. In recent studies, a new biofuel source has been screened from fungi; the high growth rate of fungi makes them more applicable to biodiesel industries than plants. We believe that the carbon source in OPEM stimulates the fungus to produce more fatty acids. Furthermore, filamentous fungi serve as an ideal source of bio-oil production [27]. A study by Bhanja et al.[28] reported that bio-oil from fungi was extracted from high lipid-containing biomass by transesterification. Some studies have shown that a low ratio of nitrogen to carbon concentration in the media leads to decreased protein synthesis and facilitates an increase in lipid production [29].”

Lines 182-185: Wrong conclusion: The fact that “Eudistomin I is a B-carboline alkaloid, an indole alkaloid class that has been isolated from the marine Caribbean ascidian Eudistoma olivaceum” does NOT suggest that “B-carboline alkaloid is biosynthetically derived from tryptamine.”

Methods: hierarchical clustering description is missing

Chaotic flow: the figures appear in a strange order. I would assume that first co-culture was grown and then compounds were fractionated/isolated; Figure 8 is mentioned before Figure 7.

Inappropriate use of word: just in the Introduction:”successfully” (line 50); “fractions” (line 68); “isolated” (line 68); “Post-acquisition treatment data were analysed” (line 73); “extracted” and “freely” (line 75); “identification and elucidation” (line 78))

Unclear sentence (lines 359-360): The positive results of the fractionation were subjected to LC-TOF MS for multivariate statistical data analysis and metabolite identification.

What is it?/What is the meaning of…?/potential typo

Line 88: What are G0, G3 and G5 co-cultures?

Line 97: What is the meaning of “*G0, XP and G.boninense is co-culturing together at the same time”?

Fig 2.: What is “recycling” HPLC?

Line 113: What is “X matrix” or “X variable”?

Line 121: To maximize the difference of metabolites between groups, the PLS-DA model (figure 4) was then applied. What is the meaning of “maximize the difference”?

Fig 3.: What is “control-opem”?

Lines 137-139: Looking at the samples with positive scores in PC1 in this model, the control OPEM masses were grouped (positive score) due to the presence of 133.048 m/z at a retention time of 2.11 min, which was separated from the treatment group (figure 5). What is the meaning? In addition, there is no 133.48 m/z in Fig5.

Line 152: What is VIP?

Line 158: “The heatmap of the respective metabolites corresponding to each group is also presented”. What is a metabolite heatmap?

Lines 161-162: The most important metabolites (by mass) responsible for the apparent discrimination (those with VIP>1). What is the meaning?

Line 343: What is “Isolation of fractionation”?

Line 369: 5% B 15-22min?

Line374: Data Analysis?

Line390: P<0.005  ?

Author Response

Response to Reviewer 2 Comments:

Major issue:

“Metabolic profile of Scytalidium parasiticum-Ganoderma boninense co-cultures revealed the alkaloids, flavonoids and fatty acids that contribute to anti-Ganoderma activity” by Rafidah Ahmad et al., presents a study of the interactions of Scytalidium parasiticum and Ganoderma boninense, two fungi involved in basal stem rot disease of the oil palm trees. Based on previously published data it was concluded that Scytalidium parasiticum is capable of inhibiting a growth of Ganoderma boninense species, so it has become an attractive target as a new biocontrol agent. Rafidah Ahmad et al. have used a G. boninense and S. parasiticum co-cultivation approach to identify small molecules (metabolites) involved in fungi communication and specifically in regulation of the growth of the parasitic species.

I think that experimental design of this study and selected methods have a great potential to reveal valuable details of G. boninense and S. parasiticum interactions. Regrettably, the manuscript is poorly written. There are very few typos, so it looks like the effort was taken to prepare manuscript for submission. However, the text is full of grammatical errors: unclear sentences, inappropriately used words, two-sentence paragraphs (!) and future tense used to describe the results (!!). The only acceptable parts of this manuscript are the Abstract and Figures (except Fig7 which is illegible). Sadly, all figures are missing their figure legends. I found the Abstract interesting, but the Introduction is a bit disappointing since it inadequately sets the stage for the presented research. Unfortunately, the Results section contains information which would be more fitting for the figure legend/s and the Discussion contains mostly observations. In addition, the excessive use of the colloquial language sometimes gives the reader the impression of a lack basic understanding of applied statistical methods or identification tools.

Response : The manuscript has been sent to English Proof Reading Services before it is sent to this journal. Attached is the proof of English Editing service. However, we take into account of your comments and the manuscript has been sent to different English editing service to improve the language (Please find an attachment of the proof of English editing service) The correction has been made to remove the two sentences paragraph.

Even though the manuscript requires significant revisions this is not the critical issue. It looks like the entire analysis has been completed based on peak intensities from base peak chromatograms (BPC). This is a serious mistake, since the BPC represents only the intensity of the most intense peak at every point in the analysis; therefore, it has very little analytical value. Based on that I cannot consider this manuscript for publication. Below you can find a list of major/minor issues as well as the list of specific examples.

Response : Thank you for the constructive comment. In this study, BPC is used to visualize representative peaks only as BPC is less noisy than TIC. Specifically, we used profile analysis software to extract EIC (Extracted Ion Chromatogram) of each peak and used for further analysis. We have further elaborated how data analysis was conducted in materials and methods section

Other major issues to address before submission:

1.I think that an “untargeted screening approach” is a little bit of the overstatement. The extraction method imposes a bias to what kind of molecules are extracted from the co-culture. Further, additional bias is introduced by using 98% methanol to fractionate extracts. It is not a surprise that alkaloids, flavonoids and fatty acids were among the identified metabolites.

Response 1: We agreed with the comment. Therefore, the term ‘untargeted’ has been removed in the manuscript for better clarity.

2. It is unclear how the authors determined the relative concentration of identified compounds (Table3).

Response 2: The relative concentration was determined based on the peak area calculated from EIC (extracted ion chromatogram) to the standard curves obtained from internal standard (caffeic acid) from different concentration (0.1, 1, 10, 50,100 mg/ml). The relative concentration can be considered as semi-quantitative and not absolute quantitative.  We have added a new information on relative concentration in section 4.6. LC-TOF-MS Analysis

3. It is unclear how metabolites were identified. There seems to be some confusion regarding “putative” and “confident” identification. In addition, I find it hard to believe that the essential amino acids had to be identified using a standard only because there was no fragmentation data.

Response 3: Compound identification of metabolites was performed by comparing of accuracy m/z value (<20 ppm), and MS/MS spectra with an in house database, HMDB, METLIN and KEGG as well as comparison with a pure standard compound. This has been explained in section 4.8 Metabolite identification. For the amino acid, we are comparing the retention time (acceptable retention time tolerance 0.01 min) and accurate mass of precursor ions generated from EIC (acceptable mass errors <3 ppm) to a pure standard, more than 98%. According to FDA, in confirmation of compound with pure standard can be made with exact mass data from high resolution mass spectrometry (HRMS)- Time of flight (TOF). Analytes were considered identified if both retention time and accurate mass criteria were fulfilled. Nevertheless, we provide the fragmentation data to support our findings (Table 3) as we run both MS and MS/MS for the reference standard.

Minor issues to address before submission:

1. Lines 172-173: “unknown )metabolites” means that they cannot be found in any data base. Therefore, the following statement is not true: “In this study, possible metabolites that were expressed in response to G. boninense-S. parasiticum co-culture were reported for the first time.”

Response 1: We remove the sentences as it makes some confusion to the reviewer. The ‘first time’ here means, this is the first report on chemical profiling of the metabolites produced from interaction of G. boninense-S. parasiticum. The word does not mean the unknown metabolite was reported for the first time. Since this paragraph discuss on the significant changes metabolite from control and treatment, we explain more on how we identified the biomarker metabolites from PCA and PLS-DA. Please refer to Section 3.1 Metabolites significantly different in the treatment group and control group.

2. Lines 207-215: Irrelevant information to manuscript scope: “Tanimura reported that microorganisms with more than 20% fatty acids as their dry weights could be considered oleaginous microorganisms [26]. In recent studies, a new biofuel source has been screened from fungi; the high growth rate of fungi makes them more applicable to biodiesel industries than plants. We believe that the carbon source in OPEM stimulates the fungus to produce more fatty acids. Furthermore, filamentous fungi serve as an ideal source of bio-oil production [27]. A study by Bhanja et al.[28] reported that bio-oil from fungi was extracted from high lipid-containing biomass by transesterification. Some studies have shown that a low ratio of nitrogen to carbon concentration in the media leads to decreased protein synthesis and facilitates an increase in lipid production [29].”

Response 2: We have remove the sentences accordingly.

3. Lines 182-185: Wrong conclusion: The fact that “Eudistomin I is a B-carboline alkaloid, an indole alkaloid class that has been isolated from the marine Caribbean ascidian Eudistoma olivaceum” does NOT suggest that “B-carboline alkaloid is biosynthetically derived from tryptamine.”

Response 3 : We rephrase the sentences and explain more on the biosynthesis of eudistomin from tryptamine .According to O beck and Faul, the biosynthesis of the B-carboline alkaloid, is associated with Pictet-Spengler reaction between indoleamines (Tryptamin and serotonin) . Other studies showed that the synthesis of eudistomin began with tryptamine as a starting material (Rogero, CM et al. 2014). Please refer to Section 3.1 Metabolites significantly different in the treatment group and control group.

4.Methods: hierarchical clustering description is missing

Response 4: We have mentioned about the hierarchical clustering in 4.9 Statistical analysis.  

5.Chaotic flow: the figures appear in a strange order. I would assume that first co-culture was grown and then compounds were fractionated/isolated; Figure 8 is mentioned before Figure 7.

Response 5: The co-cultures were grown, extracted and fractionated using HPLC recycle system. The fractions were collected and tested for anti-Ganoderma experiment. The fractions that have positive antimicrobial result will be sent for LC-TOF-MS analysis. We have change the flow, where we mentioned figure 7 (now figure 3) in the results in section 2.4 VIP list and for figure 8 (now figure 4), we also mentioned this in 2.5 hierarchical clustering analysis of metabolites. We have rearrange 2.4 and 2.5 results.

6.Inappropriate use of word: just in the Introduction:”successfully” (line 50); “fractions” (line 68); “isolated” (line 68); “Post-acquisition treatment data were analysed” (line 73); “extracted” and “freely” (line 75); “identification and elucidation” (line 78))

Response 6:The correction has been made as suggested by reviewer.

7. Unclear sentence (lines 359-360): The positive results of the fractionation were subjected to LC-TOF MS for multivariate statistical data analysis and metabolite identification.

Response 7: The correction has been made. The fraction that showed positive inhibition against Ganoderma was sent to LC-TOF-MS analysis to identify the active compound(s). Please refer to 4.5 Anti-Ganoderma activity in laboratory.

8.What is it?/What is the meaning of…?/potential typo

Line 88: What are G0, G3 and G5 co-cultures?

Response : G0, G3 and G5 are co-cultures samples according to the growth age of Ganoderma. The G0, G3 and G5 have been explained in Figure 7 (schematic diagram of the study), section 4.4.2 experimental design of culture conditions and in the footnote in Table 1.

Line 97: What is the meaning of “*G0, XP and G.boninense is co-culturing together at the same time”?

Response: G0 means S.parasiticum is grown together with G.boninense simultaneously in the same flask. G3 and G5 mean, S.parasiticum is grown to the same flask with G.boninense according to G.boninense growth age (3 days or 5 days) respectively.

Fig 2.: What is “recycling” HPLC?

Response : The information on recycling preparative HPLC was added in the introduction section, Line no 72-76. Briefly, the recycling HPLC is an advancement technology to increase separation efficiency and keeping the peak dispersion by recycling the sample into the column. It has recycled valve in the HPLC system to recirculate the unresolved peaks into the column and offers higher resolution. The eluent can be recycled to the column up to 20 times until the peak resolved and fraction collected.

Line 113: What is “X matrix” or “X variable”?

Response : The terms of X matrix and X variables are applied in the PCA and PLS-DA analysis. Principal component analysis (PCA)  is a supervised classification algorithm that is widely used for multivariate data analysis in comparing, discriminating, and classifying mass spectral data. In a typical data matrix used for multivariate statistical analysis, each row represents a different sample while the metabolite identities, m/z, or peak variables are aligned into specific columns. X matrix is the peak intensities from EIC. Y matrix is the class label (treatment and control). We clarify the term used in the manuscript, please refer to 2.2. Multivariate analysis of the G. boninense-S. parasiticum co-culture

Line 121: To maximize the difference of metabolites between groups, the PLS-DA model (figure 4) was then applied. What is the meaning of “maximize the difference”?

Response : In this multivariate analysis, we use PCA (unsupervised data) and PLS-DA (supervise data). The unsupervised data means the method does not use class label information (which group does each sample in the data table belong). The PCA model highlights differences between specific groups of samples that we are interested in, but this is not always the case. This is because PCA does not specifically take group-information into account. The method focuses on differences between samples. In other words, it may not focus on differences between groups of samples when the differences between samples within a group are much larger. Hence to focus the difference between a group, we focus on PLS-DA. In PLS-DA, we use class label information, thus PLS-DA will highlight the difference between groups. For this study, PCA has showed good separation thus we removed the PLS-DA analysis as suggested by another reviewer. We only performed the PLS-DA analysis to generate the VIP list of metabolites.

Fig 3.: What is “control-opem”?

Response: We use OPEM media as control. We will change the term as control as it makes some confusion to the reader.

Lines 137-139: Looking at the samples with positive scores in PC1 in this model, the control OPEM masses were grouped (positive score) due to the presence of 133.048 m/z at a retention time of 2.11 min, which was separated from the treatment group (figure 5). What is the meaning? In addition, there is no 133.48 m/z in Fig5.

Response: The mass of 133.048 is circle in figure 5 (now Figure 2). We try to give the best picture as we could as the mass is redundant with the other masses. The loading plot in figure 2 is correlates with figure 1, scatter plot. The reason why the control is separated from the treatment in scatter plot is because the presence of 133.048, clustered away in loading plot. For example the treatment group (red triangle) is separated from the other group due to the presence of 288.284 m/z. Please refer to attachment for further explanation

Line 152: What is VIP?

Response: We have added a statement on VIP for better understanding in 2.4. Variable Influence on Projection (VIP) List. Briefly, variable influence on projection (VIP) is commonly used to summarize the importance of the X-variables in multivariate models based on projections. It summarizes the contribution a variable makes to the model. The value of VIP score which is greater than 1 is the typical rule for selecting relevant variables. The VIP can only be generated from PLS-DA analysis.

Line 158: “The heatmap of the respective metabolites corresponding to each group is also presented”. What is a metabolite heatmap?

Response : We have added the statement on heat map for better clarification. Please refer to 2.5. Hierarchical Clustering Analysis of Metabolites. A heat map is a data visualization technique that shows magnitude of a phenomenon as color in two dimensions. The variation in color is due to intensity of the compound based on EIC. Row display metabolites and column display samples.

Lines 161-162: The most important metabolites (by mass) responsible for the apparent discrimination (those with VIP>1). What is the meaning?

Response: Brief explanation for VIP has been added in 2.4. Variable Influence on Projection (VIP) List .  In the component matrix, variables with higher value indicate a higher contribution of discrimination from groups of that component.

Line 343: What is “Isolation of fractionation”?

Response : The sentence has been changed to fractionation of sample. Please refer to 4.4.4.  Recycling Preparative HPLC Fractionation of Sample

Line 369: 5% B 15-22min?

Response: The correction has been made accordingly in 4.6 LC-TOF-MS Analysis

Line374: Data Analysis?

Response: Data Analysis 4.0 is a software use for mass spectral data.

Line390: P<0.005 ?

We choose p value less than 0.005 from ANOVA analysis for significant differences between treatment and control

Reviewer 3 Report

The paper addresses an interesting area, biological control of basal stem rot. It investigates this by metabolic profiling, which is also interesting.

At present, however, the study is not very accessable, as the presentation of the results and the methods need to be distinctly improved.

Suggestions/comments

Fig. 1 should be removed and replaced by a schematic overview of the study design. Further, the methods section should explain the study design in more detail. It was not clear to me what "As shown in figure 1, the positive results of the G0 fractions were observed in the second, third and fifth recycles" means. What is a recycle? What is a G3 co-culture?

The example agar plate is not clear to me, either. To me, it looks like "G0 R3 37-41" and the negative control are identical, but for the former on inhibition zone is noted. Can the authors explain or provide a better example?

Figs 3 and 4 are identical, even though PCA and PLS-DA are very different analyses. Given the PCA achieves good separation, there is not need for a PLS-DA.

Fig 5 is a PCA loadings plots, not a 'PCA of a loadings plot'. 

Fig 7 is not readable to me due to being a small labels bitmap. It should a vector file.

From what I gather, the authors tested preparative fractions against Ganoderma. However, they report two sets of retention times, the preparative fraction times and the RT from the subsequent run to further analyse the fractions. Am I getting the wrong end of the stick here? The latter times are not really relevant and should appear in the appendix if at all.

Author Response

The paper addresses an interesting area, biological control of basal stem rot. It investigates this by metabolic profiling, which is also interesting.

At present, however, the study is not very accessable, as the presentation of the results and the methods need to be distinctly improved.

Response: We provide separate file for picture to make it more clear.

Suggestions/comments

Fig. 1 should be removed and replaced by a schematic overview of the study design. Further, the methods section should explain the study design in more detail. It was not clear to me what "As shown in figure 1, the positive results of the G0 fractions were observed in the second, third and fifth recycles" means. What is a recycle? What is a G3 co-culture?

Response : We remove Figure 1 as supplementary materials and we have included the schematic overview of the study as requested (Figure 7). The 2nd, 3rd  recycle means the eluent has repeatedly recycle into the same column to achieve separation before fraction. Each recycle of sample improved resolution and separation (J. Sidana and L. K. Joshi, 2013). The terms of G3 co-culture has been explained in 4.4.2. Experimental design of Culture Conditions.

The example agar plate is not clear to me, either. To me, it looks like "G0 R3 37-41" and the negative control are identical, but for the former on inhibition zone is noted. Can the authors explain or provide a better example?

Response : The figure is merely an additional information. Hence, we have decided to remove it from the text. The actual data of inhibition zone was shown in Table 1.

Figs 3 and 4 are identical, even though PCA and PLS-DA are very different analyses. Given the PCA achieves good separation, there is not need for a PLS-DA.

Response: Indeed the figure 3 and figure 4 looks identical although the analysis are different, PCA and PLS-DA. This is because PCA has showed a good separation, so PLS-DA achieves similar pattern to PCA. We have remove the PLS-DA analysis as suggested as PCA gives better separation. The reason that we proceed the analysis to PLS-DA is to generate the list of VIP metabolites that significantly project the changes. This has been mentioned in the text section 2.4 in the VIP list.

Fig 5 is a PCA loadings plots, not a 'PCA of a loadings plot'. 

Response: We have made the correction accordingly

Fig 7 is not readable to me due to being a small labels bitmap. It should a vector file.

Response: We have attached figure 7 (now figure 3) as separate file

From what I gather, the authors tested preparative fractions against Ganoderma. However, they report two sets of retention times, the preparative fraction times and the RT from the subsequent run to further analyse the fractions. Am I getting the wrong end of the stick here? The latter times are not really relevant and should appear in the appendix if at all.

Response: We have include the schematic diagram (Figure 7) of the study give an overview of our study. Yes, two RTs have been mentioned in the text (RT from Recycling preparative HPLC) and RT of LC-TOF-MS analysis. The fractions were collected based on peak detected by UV wavelength that appeared at certain RT in recycling preparative HPLC. The fractions than were tested for antimicrobial activity and later sent for LC-TOF-MS analysis. In the LC-TOF-MS results, the RT was stated for each peak in the chromatogram. The RT in LC-TOF-MS is important to confirm the identity of the metabolites with reference standard, therefore it is relevant to show it in table 3

Round 2

Reviewer 1 Report

The authors have made the changes suggested to them! This study tells a nice and concise story about the role of secondary metabolites in plant protection against the fungus, and is ready to be shared with readers.

Author Response

We would like to thank for your constructive comments and suggestions which have allowed us to improve the clarity of this manuscript. We appreciate your time and effort which gives us the opportunity to enable us to publish this study. 

Reviewer 2 Report

I would like to thank the authors for taking the time to answer my questions and concerns in such detail.  The editorial effort has significantly improved the clarity of the presented research.  However, it has also opened the door for more questions. One of the major concerns I had previously was about the metabolite concentrations. I agree that mass spectrometry is only a semi-quantitative method, however for a different reason. Since the ability to ionize, ion transfer through the mass analyzer and finally the detection is different for each molecule, the relative metabolite concentration cannot be calculated based on a common reference. Since it appears that the difference in metabolite abundance between samples was calculated based on normalized peak intensity, please remove concentration values from Table 3. It is great to hear that data analysis was completed on the entire data set. However, if the BPC was used for visualization purposes only, why was the area under the peak calculated (see Fig 5)? This is just confusing. If this is going to be included in the manuscript, please provide a high-resolution figure presenting the BPC in place of the low quality screenshot. Also the figure legend is missing (what do these peaks represent?). In addition, I have found the new description of applied statistical methods confusing. How many times were the data sets normalized before multivariate analysis? It looks like first the EICs were normalized to total intensity (see ref.39), then pareto scaled and finally autoscaled (current section 4.7). Please clarify that. Also, the heat map is not a statistical analysis (current line 531). Finally, most of the figures contain only figure title or an incomplete legend, for example, what is the green circle in Fig2? Current Fig3 is still illegible. Fig7 is a great addition since it offers a brief overview of the applied methodology. Therefore, this should be Fig1 rather than Fig7. There is still a two-sentence paragraph (lines 53-57).

Author Response

I would like to thank the authors for taking the time to answer my questions and concerns in such detail.  The editorial effort has significantly improved the clarity of the presented research.  However, it has also opened the door for more questions. One of the major concerns I had previously was about the metabolite concentrations. I agree that mass spectrometry is only a semi-quantitative method, however for a different reason. Since the ability to ionize, ion transfer through the mass analyzer and finally the detection is different for each molecule, the relative metabolite concentration cannot be calculated based on a common reference. Since it appears that the difference in metabolite abundance between samples was calculated based on normalized peak intensity, please remove concentration values from Table 3. It is great to hear that data analysis was completed on the entire data set. However, if the BPC was used for visualization purposes only, why was the area under the peak calculated (see Fig 5)? This is just confusing. If this is going to be included in the manuscript, please provide a high-resolution figure presenting the BPC in place of the low quality screenshot. Also the figure legend is missing (what do these peaks represent?)

Response : Thank you for the details explanation on why relative concentration is not valid if calculated based on one common reference. We agree with your statement, therefore, we have removed the relative concentration in Table 3 and also in discussion and method sections. Yes, ideally  we provided the BPC for the visualize purpose only. The ion intensities in heat map (figure 5) and bar chart (figure 4) for all the metabolites are  based on EIC generated from profile analysis software. As we agree with your comment, we have replaced the BPC image to a higher resolution image as we could, eventhough the original picture has a low quality image extract from Data Analysis Software. We have rephrased the figure 5 caption for details explanation.

In addition, I have found the new description of applied statistical methods confusing. How many times were the data sets normalized before multivariate analysis? It looks like first the EICs were normalized to total intensity (see ref.39), then pareto scaled and finally autoscaled (current section 4.7). Please clarify that.

Response : It is true that first EIC was normalized to total intensities according to reference 39 in advanced bucketing.  The pre-processing data is done by using profile analysis software as this is a crucial step before applying to mutivariate statistical (PCA) analysis. The method is also applied to HCA and heat map analysis as well. There is an option of doing the PCA analysis with other software such as SIMCA-P or continued with profile analysis software that has been used in reference 39. In this case we are choosing SIMCA-P for multivariate analysis (PCA) instead of continued using the profile analysis software. The reason behind this is to perform the pre-treatment methods. we take more effort in improving our biological  information in our dataset as to use a few pre-treatment methods (pareto scaling and log-transformed).

Pareto scaling is used as it enhances the contribution of lower concentration metabolites without amplifying noise and artifacts present in metabolomic dataset (Cloarec, 2005) and log transformed makes the data distribution normally distributed and stabilize the variance. It removes the heteroscedasticity of the data.  In metabolomic research, different data pre-processing steps are applied to generate a good and reliable data in the form of  normalized EIC.  They were found to greatly affect the outcome of the data analysis. Moreover, selecting a proper data pretreatment method is an essential step in the analysis of metabolomics data and greatly affects the metabolites that are identified to be the most important. (van den Berg, R.A et al. 2006). Although this paper used GC-MS analysis, the pre-treatment method has been vastly used in LC-MS as well (Chanana et al. 2017). We have removed autoscale, as we used pareto scale in the analysis.

Also, the heat map is not a statistical analysis (current line 531).

Response: We have moved the heat map and HCA explanation in section 4.7. Data processing and Data Analysis

Finally, most of the figures contain only figure title or an incomplete legend, for example, what is the green circle in Fig2? Current Fig3 is still illegible. Fig7 is a great addition since it offers a brief overview of the applied methodology. Therefore, this should be Fig1 rather than Fig7.

Response: We have rephrased all the figures caption for Figure 1, Figure 2, Figure 3, Figure 4, Figure 5, Figure 6 and Figure 7  in details. We provided higher resolution image for figure 6 (we try our best to increase the resolution) and also figure 3( we attached separate file image- the image in the manuscript has been compressed to suit the word processing software, that's why it is illegible). We changed the figure sequence as suggested. Figure 7 has been renumbered  to Figure 1. We have explained the green dot in figure 2 (now figure 3) as below:

PCA loading plot of control and treated samples that showed the distribution of metabolites (masses) in wc1 and wc2 planes (green dots). The most important metabolites in VIP list are highlighted in red. Circle in blue is metabolite with 133.048 m/z (Tryptamine)

There is still a two-sentence paragraph (lines 53-57).

Response: The two- sentences paragraph has been removed and combined with previous paragraph.

Reviewer 3 Report

The authors did respond scholarly to my comments and have - in my opionion - improved the manuscript. It is a bit hard to follow all changes through the tracked changes document, but overall clarity has improved.

I think this manuscript can now be published.

Author Response

We would like to thank for your constructive comments and suggestions which have allowed us to improve the clarity of this manuscript. We appreciate your time and effort which gives us the opportunity to publish this study. 

This manuscript is a resubmission of an earlier submission. The following is a list of the peer review reports and author responses from that submission.